OBSERVATION

# Breaching the Barrier: Genome-Wide Investigation into the Role of a Primary Amine in Promoting *E. coli* Outer-Membrane Passage and Growth Inhibition by Ampicillin

Claire Maher,[a,b] Ram Maharjan,[b,c] Geraldine Sullivan,[b,c] Amy K. Cain,[b,c] Karl A. Hassan[a,b]

[a]College of Engineering, Science and Environment, University of Newcastle, Callaghan, New South Wales, Australia
[b]ARC Centre of Excellence in Synthetic Biology, Macquarie University, Macquarie Park, New South Wales, Australia
[c]School of Natural Sciences, Macquarie University, Macquarie Park, New South Wales, Australia

**ABSTRACT**    Gram-negative bacteria are problematic for antibiotic development due to the low permeability of their cell envelopes. To rationally design new antibiotics capable of breaching this barrier, more information is required about the specific components of the cell envelope that prevent the passage of compounds with different physiochemical properties. Ampicillin and benzylpenicillin are $\beta$-lactam antibiotics with identical chemical structures except for a clever synthetic addition of a primary amine group in ampicillin, which promotes its accumulation in Gram-negatives. Previous work showed that ampicillin is better able to pass through the outer membrane porin OmpF in *Escherichia coli* compared to benzylpenicillin. It is not known, however, how the primary amine may affect interaction with other cell envelope components. This study applied TraDIS to identify genes that affect *E. coli* fitness in the presence of equivalent subinhibitory concentrations of ampicillin and benzylpenicillin, with a focus on the cell envelope. Insertions that compromised the outer membrane, particularly the lipopolysaccharide layer, were found to decrease fitness under benzylpenicillin exposure, but had less effect on fitness under ampicillin treatment. These results align with expectations if benzylpenicillin is poorly able to pass through porins. Disruption of genes encoding the AcrAB-TolC efflux system were detrimental to survival under both antibiotics, but particularly ampicillin. Indeed, insertions in these genes and regulators of *acrAB-tolC* expression were differentially selected under ampicillin treatment to a greater extent than insertions in *ompF*. These results suggest that maintaining ampicillin efflux may be more significant to *E. coli* survival than full inhibition of OmpF-mediated uptake.

**IMPORTANCE**    Due to the growing antibiotic resistance crisis, there is a critical need to develop new antibiotics, particularly compounds capable of targeting high-priority antibiotic-resistant Gram-negative pathogens. In order to develop new compounds capable of overcoming resistance a greater understanding of how Gram-negative bacteria are able to prevent the uptake and accumulation of many antibiotics is required. This study used a novel genome wide approach to investigate the significance of a primary amine group as a chemical feature that promotes the uptake and accumulation of compounds in the Gram-negative model organism *Escherichia coli*. The results support previous biochemical observations that the primary amine promotes passage through the outer membrane porin OmpF, but also highlight active efflux as a major resistance factor.

**KEYWORDS**    efflux pumps, Gram-negative bacteria, outer membrane

Gram-negative pathogens are naturally resistant to many compounds that effectively inhibit Gram-positive bacteria due to the low intrinsic permeability of their cell envelopes, making them a high priority for new antibiotic development (1). The Gram-negative cell envelope is complex. It notably comprises two membranes with distinct chemical characteristics

Address correspondence to Karl A. Hassan, karl.hassan@newcastle.edu.au.

The authors declare no conflict of interest.

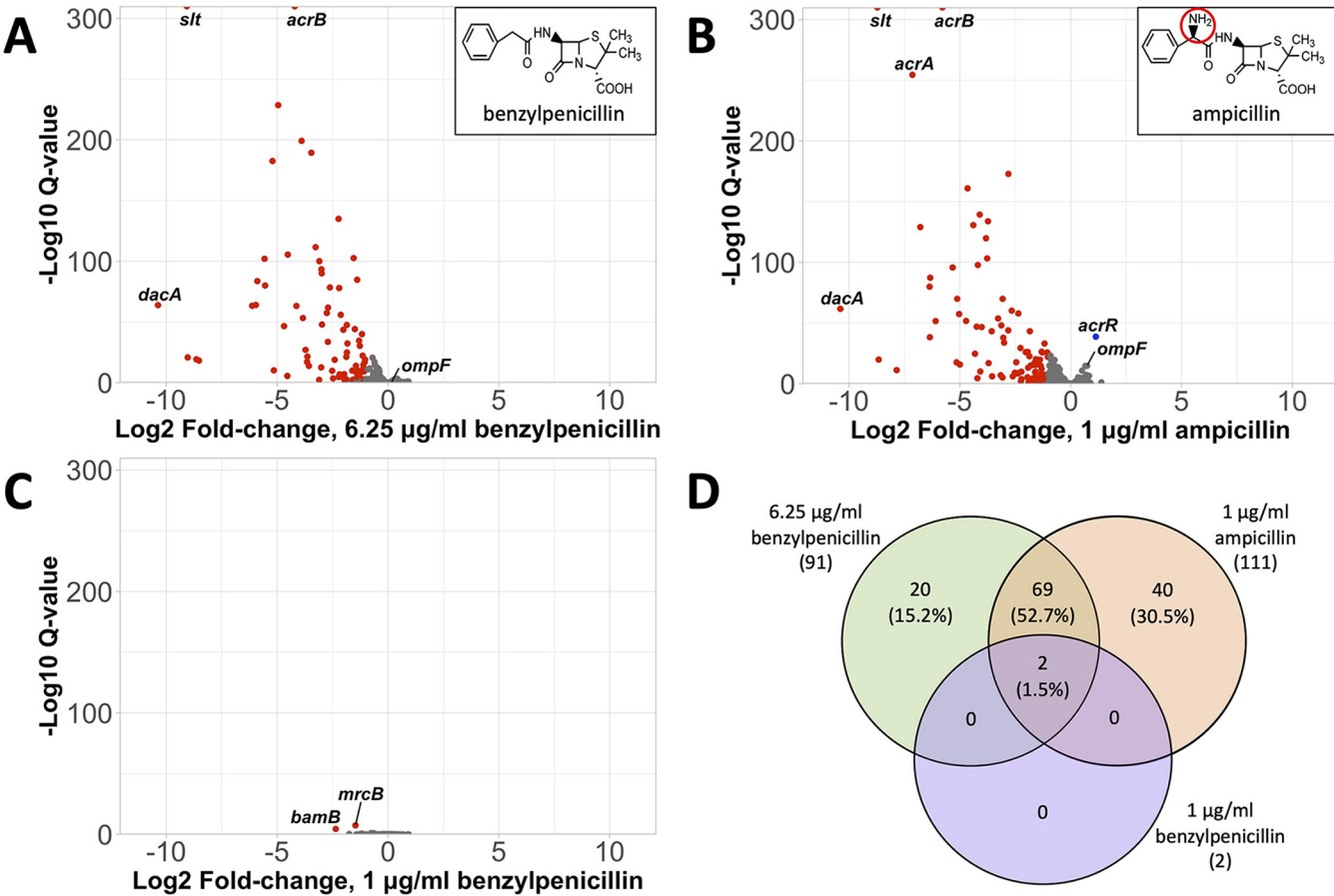

**FIG 1** Challenging BW25113 transposon library with subinhibitory concentrations of ampicillin and benzylpenicillin. Volcano plots of the fold change in transposon insertions in genes between antibiotic challenges and the untreated control are shown for 6.25 μg/mL PEN (A), 1 μg/mL AMP (B), and 1 μg/mL PEN challenges (C). Red datapoints are insertion changes with a ≥2-fold decrease and q.value <0.05, blue datapoints are insertion changes with a ≥2-fold increase and q.value <0.05, and gray datapoints are insertion changes with q.value ≥0.05. Inserts show the chemical structures for PEN (A) and AMP (B), with the primary amine group circled in red. (D) Venn diagram of overlap between genes with significant changes in insertion numbers in different challenge conditions.

and a network of active efflux pumps that typically display broad substrate specificities. The Gram-negative inner membrane is composed of phospholipids, whereas the outer membrane is polar, composed of a phospholipid inner leaflet and an outer leaflet of lipid A decorated with complex polysaccharides on its outer surface that vary considerably between strains and species. Detailed information linking specific components of the Gram-negative cell envelope to the exclusion or uptake of compounds with distinct physiochemical properties would aid in the rational design of new antibiotics that can penetrate and stay within Gram-negative cells. This study investigated the importance of a primary amine group in promoting the passage of β-lactam antibiotics in the model Gram-negative species *Escherichia coli* using a nontargeted genomic approach.

Ampicillin (AMP) and benzylpenicillin (PEN) are identical except for the presence of a primary amine group in AMP that is absent in PEN (Fig. 1, insets to panels A and B). This chemical feature was developed in the 1960s to improve PEN which is only clinically effective against Gram-positive species (2, 3). The primary amine has been identified as promoting compound accumulation in Gram-negatives (4), providing AMP with broad-spectrum antibiotic activity. It is known from genetic and biochemical studies that the primary amine in AMP improves its uptake through outer membrane (OM) porins, such as OmpF in *E. coli* (5, 6). However, other specific factors that may promote or impede the passage of β-lactam antibiotics across the Gram-negative cell envelope have not been comprehensively explored. Here, we applied Transposon Directed Insertion-site Sequencing (TraDIS) to identify the parts of the Gram-negative cell envelope that act to exclude or facilitate uptake of AMP and PEN, and thus infer their pathways of accumulation and the importance of the primary amine in AMP to overcome this

barrier. More generally, this information will highlight specific elements of the barrier that could be targeted to potentiate compound entry.

TraDIS was performed using a high-density transposon mutant library of *E. coli* K-12 strain BW25113 (>400,000 unique mutants), constructed using a custom EzTn*5* transposome according to previously described protocols (7, 8). The library was challenged with equivalent subinhibitory concentrations of AMP and PEN that resulted in near identical growth perturbations in the parental strain; 1 $\mu$g/mL AMP and 6.25 $\mu$g/mL PEN (~20% growth inhibition as determined by liquid media growth curves in Miller LB media), and the mutant profiles in the challenged libraries were compared to untreated controls using the TraDIS toolkit (9). Cells were sampled at stationary phase to ensure the effects of compound accumulation throughout the cell cycle, including during active growth phases, were reflected in mutant abundance (10). As AMP and PEN have the same antibacterial targets and modes of action (11), it was expected that considerable overlap between the significantly differentially selected genes would exist between the two treatment conditions, particularly for genes encoding their antibiotic targets and downstream systems that may compensate for target inhibition. Differences in mutant selection between the conditions could reflect alternative pathways of compound accumulation. An additional treatment of 1 $\mu$g/mL PEN was also included as a control to match AMP by media concentration.

Genes identified as important during antibiotic exposure were classified as having a +/-≥2-fold change in mutant abundance (as measured by change in relative transposon insertion frequency) between antibiotic-treated cells and untreated controls and q-value < 0.05. There were 111 such genes following 1 $\mu$g/mL AMP treatment, 91 following 6.25 $\mu$g/mL PEN treatment, and only 2 following 1 $\mu$g/mL PEN treatment (Fig. 1A,B; Data set S1). With the exception of *acrR* in the 1 $\mu$g/mL AMP treatment, all of these genes had a decrease in insertions following treatment, indicating that their disruption was deleterious to *E. coli* fitness under the conditions tested. The two significantly differentially selected genes in the 1 $\mu$g/mL PEN treatment were *mrcB* which encodes a bifunctional DD-transpeptidase/glycosyltransferase class 1 penicillin binding protein (PBP1A), a potential target of PEN, and *bamB*, encoding part of the $\beta$-barrel assembly machinery (BAM) complex, highlighting the importance of $\beta$-barrel proteins in the outer membrane and/or their products for the exclusion of PEN (Fig. 1C). Both genes were also significantly differentially selected under the two other antibiotic conditions tested.

**Selection against insertions in $\beta$-lactam targets.** Approximately 54% of the significantly impacted genes were shared between the 1 $\mu$g/mL AMP and 6.25 $\mu$g/mL PEN treatments (Fig. 1D). To identify functional trends, the genes were assigned cluster of orthologous groups (COG) categories (Fig. 2). The most highly represented COG category was cell wall/membrane/envelope biogenesis (M; Fig. 2). Many of the genes differentially selected by both antibiotics in this category involved in cell wall synthesis, including several penicillin binding proteins (PBPs), the targets of $\beta$-lactams. Indeed, mutations in *dacA*, encoding the major carboxypeptidase in *E. coli* PBP5, were the most highly selected against of any gene in both treatments. It has been suggested that PBP5, a redundant PBP, may act to sequester $\beta$-lactams away from essential PBPs, providing a shielding effect (12). This may explain why insertions in this PBP in particular were so detrimental to fitness under both treatments.

Insertions in other nonessential PBP genes such as *mrcA* and *mrcB*, encoding PBP1A and PBP1B, respectively, also had similar impacts on cell fitness under both the 1 $\mu$g/mL AMP and 6.25 $\mu$g/mL PEN treatments (although insertions in *mrcA* fell below the 2-fold threshold). *lpoB*, encoding an activator of McrB, was among the genes with the greatest decrease in insertions for either treatment. Insertions in several other genes involved in cell envelope production were also selected against by both treatments at similar levels, such as *slt*, which encodes a lytic transglycosylase thought to be involved in peptidoglycan quality control. Thus, many of the genes whose disruption most greatly decreased fitness under both antibiotics were either PBPs or genes that may compensate for PBP target inhibition, such as *slt*. The highly similar levels of selection against disruption of these genes supports the equivalence of the AMP and PEN concentrations tested.

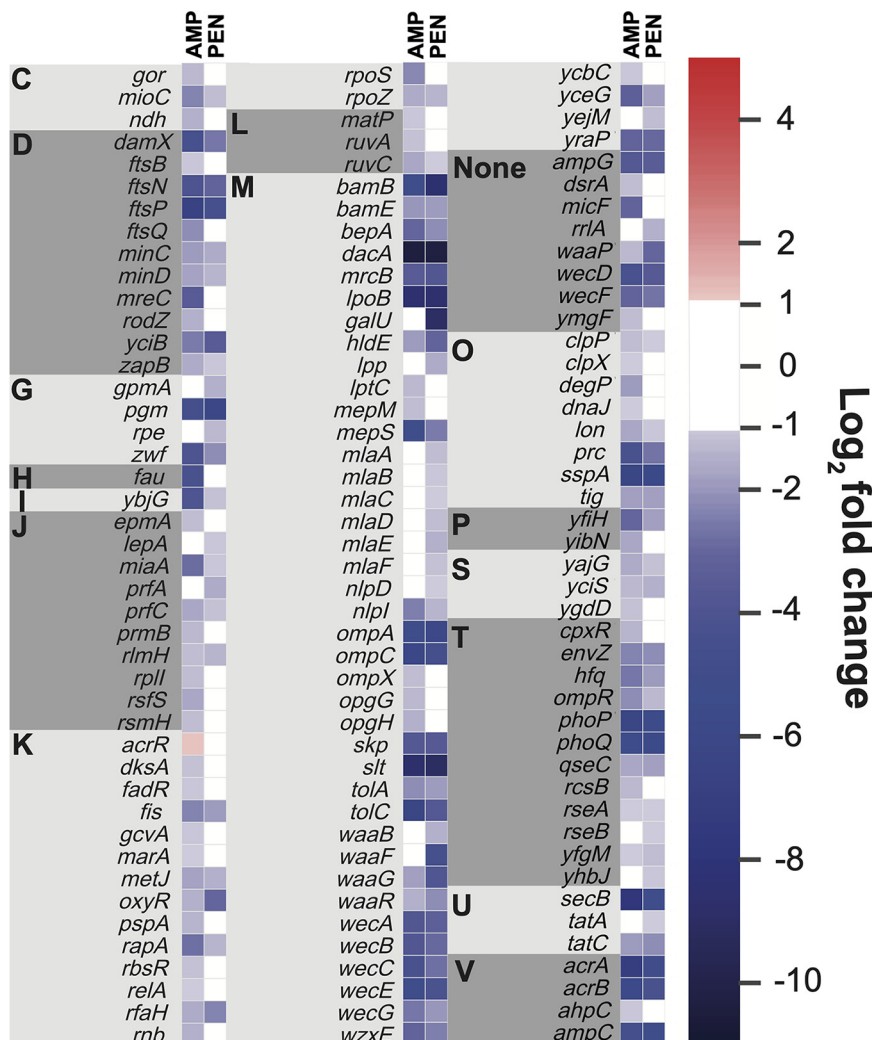

**FIG 2** Insertion frequency changes following ampicillin and benzylpenicillin treatment. Transposon insertion site numbers within each gene are presented as $\log_2$(FC), compared to the control group for ampicillin (AMP) and benzylpenicillin (PEN). Only genes with a q.value <0.05 and a fold change ≥abs(1) are shown. Genes are functionally grouped according to their COG categories: [C] Energy production and conversion, [D] Cell cycle control, cell division, chromosome partitioning, [G] Carbohydrate transport and metabolism, [H] Coenzyme transport and metabolism, [I] Lipid transport and metabolism, [J] Translation, ribosomal structure and biogenesis, [K] Transcription, [L] Replication, recombination and repair, [M] Cell wall/membrane/envelope biogenesis, [O] Posttranslational modification, protein turnover, and chaperones, [P] Inorganic ion transport and metabolism, [S] Function unknown, [T] Signal transduction mechanisms, [U] Intracellular trafficking, secretion, and vesicular transport, [V] Defense mechanisms.

**Selection against insertions in LPS-related genes under benzylpenicillin treatment.** 60% of genes uniquely affected by PEN treatment fell into the cell wall/membrane/envelope biogenesis COG category (M; Fig. 2). These genes were primarily involved in LPS synthesis and maintenance, such as *yejM*, which is implicated in lipid homeostasis; *lpp*, which anchors the OM to the cell wall; all genes making up the *mla* pathway (*mlaA-F*), which plays a role in maintaining LPS asymmetry; and the LPS core biosynthesis genes *waaF* and *waaB*. Although not uniquely affected, *waaG* and *waaP* were also both more negatively selected under PEN than AMP treatment. Mutations in both the *waa* and *mla* pathways compromise the OM and can significantly increase OM permeability and sensitivity to hydrophobic antibiotics like PEN (13, 14).

Another gene uniquely affected by PEN treatment was *galU*, which had one of the greatest decreases in insertions under PEN treatment. To examine the importance of *galU* for PEN tolerance, a targeted mutant (15) was examined. Deletion of *galU* resulted in a 2-fold reduction in PEN MIC (Fig. S1). GalU is involved in the production of UDP-d-glucose, a central

precursor used in the synthesis of several OM components, including LPS, and is also essential in the galactose degradation pathway. Deletion of *galU* results in a compromised OM, as well as reduced levels of TolC (16), which may explain the impact of insertions in *galU* on cell fitness under PEN treatment.

The prominence of OM and particularly LPS-related genes among those most highly and, in some cases, uniquely detrimental to fitness under PEN treatment suggests that the LPS layer is crucial in preventing the uptake of PEN, aligning with the current understanding of poor PEN passage through OM porins in *E. coli*.

Functional trends among significantly affected genes were less apparent under AMP than PEN treatment. Overall, genes for which insertions were more negatively selected compared to PEN appeared to be involved in general stress responses rather than compound uptake. These included transcriptional regulators such as *rpoS*, which encodes the central regulator of the general stress response in *E. coli* and which has previously been observed to play a role in AMP resistance (17).

Another transcriptional regulator, *acrR*, which encodes a repressor of the AcrAB-TolC efflux system, had a significant increase in insertions under AMP treatment, and was the only gene with a significant increase in insertions >2-fold in any of the treatments tested according to the threshold used. Conversely, mutations in *marA* which encodes an activator of *acrAB* expression were negatively selected by AMP but not PEN treatment (Data set S1). The expression of *marA* is controlled by MarR in response to antibiotic and other stresses (18, 19). Insertions in the genes encoding AcrAB-TolC were all negatively selected under both AMP and PEN treatments (Fig. 2), albeit slightly more highly under AMP treatment, indicating that this system is important for the efflux of both compounds. This was supported by MIC assays which showed a 2-fold decrease in PEN MIC in an *acrA* deletion strain, compared to a 4-fold decrease in AMP MIC (Fig. S1).

Previous work has suggested that PEN may be a better substrate of AcrAB-TolC in *E. coli* than AMP (20). This may explain why AMP would impose higher selective pressure for the inactivation of the *acrR* repressor gene and maintenance of the *marA* activator gene than PEN, as a higher level of *acrAB* expression may compensate for lower levels of AMP transport by this pump. Alternatively, PEN may be better able to promote *acrAB* expression than AMP and thus reduce selective pressure for mutations that increase its expression. Indeed, PEN has a higher membrane permeability than AMP and thus greater potential to diffuse into the cytoplasm where it could interact with cytoplasmic regulators like AcrR to relieve translational repression of *acrAB* (20, 21). To investigate this possibility, qRT-PCR was performed to examine *acrAB* expression in wild-type cells treated with AMP and PEN at concentrations equivalent to those used in this assay (22). These experiments did not show a clear change in expression for these genes for either antibiotic treatment (data not shown). Another possible reason for selective pressure on *acrR* and *marA* under AMP selection could be the roles of these regulators in controlling other genes, such as *ompF*.

The porin OmpF has previously been shown to be important for the uptake of ampicillin into the *E. coli* periplasm. The *ompF* gene had only a slight increase in insertions under AMP treatment (less than 2-fold) and no significant change under PEN treatment. These results are consistent with the greater capacity of AMP to pass through OmpF compared to PEN, but suggest that under the conditions tested maintaining AMP efflux via AcrAB-TolC is more beneficial to cells than disrupting uptake via OmpF. Due to the role of OmpF in mediating the OM passage of other small molecules important to cell fitness, its inactivation may have an overall negative fitness effect. This is supported by the decreased growth rate of BW25113 Δ*ompF* compared to wild-type (Fig. S2). However, there is evidence of selection for reduced *ompF* expression via regulatory mutations, particularly under AMP selection. For example, selection against insertions in *cpxR*, which encodes the response regulator of the CpxR/CpxA two-component regulatory system, was observed after AMP but not PEN treatment. Phosphorylated CpxR has been shown to repress expression of *ompF* and so its maintenance could reduce *ompF* expression levels (23). Conversely, mutations in the cognate sensor kinase gene *cpxA* have been shown to increase *ompF* expression in the presence of acetyl phosphate due to increased levels of phosphorylated CpxR (23). Selection for insertions in *cpxR* was observed

under AMP, but not PEN, exposure. Further, insertions in the small RNA *micF*, which inhibits *ompF* expression (24, 25), were strongly selected against only under AMP treatment. *micF* may also be repressed by AcrR and activated by MarA (26, 27). This could therefore provide an alternate explanation for selection for insertions in *acrR* and against insertions in *marA* under AMP treatment. Such insertions could lead to reduced *ompF* expression. Therefore, there may have been stronger selection for reduced expression of *ompF*, rather than its complete inactivation under AMP selection.

Previous investigations into the role of the primary amine in ampicillin accumulation in Gram-negatives have primarily focused on how this chemical feature affects compound uptake. The significance of this altered uptake efficiency in improving accumulation is supported by this study. Collectively, these results suggest that unlike PEN, compromising the LPS layer does not appear to have a drastic impact on cell survival in AMP. This is to be expected as AMP is able to enter cells through OmpF-mediated uptake. However, the potential effects of the primary amine on compound efflux and the contribution of this to changes in overall levels of accumulation have been understudied. These findings suggest that maintaining AMP efflux via AcrAB-TolC is of comparable importance to *E. coli* survival as inhibiting uptake. These results highlight the potential for TraDIS to guide investigations of antibiotic accumulation into Gram-negative bacteria.

**Data availability.** TraDIS sequence reads are available on the NCBI Sequence Read Archive and can be accessed using the NCBI BioProject accession PRJNA875563.

## SUPPLEMENTAL MATERIAL

Supplemental material is available online only.
**SUPPLEMENTAL FILE 1**, XLSX file, 0.8 MB.
**SUPPLEMENTAL FILE 2**, PDF file, 0.2 MB.

## ACKNOWLEDGMENTS

This work was supported by an Australian Research Council Future Fellowship to K.A.H (FT180100123). We thank Liam D. H. Elbourne (Macquarie University) for assistance with the Bio-Tradis bioinformatics pipeline. We thank Silas H. W. Vick (Norwegian University of Life Sciences) for assistance with heatmap figures.

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
