## [Reviewer comments · Microbiology Spectrum]

Microbiology Spectrum

Breaching the barrier: genome-wide investigation into the role of a primary amine in promoting *E. coli* outer-membrane passage and growth inhibition by ampicillin

Claire Maher, Ram Maharjan, Geraldine Sullivan, Amy Gain, and Karl Hassan

Corresponding Author(s): Karl Hassan, University of Newcastle Australia

Review Timeline:

Submission Date:	September 6, 2022
Editorial Decision:	October 14, 2022
Revision Received:	November 7, 2022
Accepted:	November 8, 2022

Editor: Philip Rather

Reviewer(s): Disclosure of reviewer identity is with reference to reviewer comments included in decision letter(s). The following individuals involved in review of your submission have agreed to reveal their identity: Jessica Mary Alice Blair (Reviewer #1)

Transaction Report:

DOI: <https://doi.org/10.1128/spectrum.03593-22>

October 14, 2022

Dr. Karl A Hassan
University of Newcastle Australia
Newcastle, NSW 2308
Australia

Re: Spectrum03593-22 (Breaching the barrier: genome-wide investigation into the role of a primary amine in promoting E. coli outer-membrane passage and growth inhibition by ampicillin)

Dear Karl,

Thank you for submitting your manuscript to Microbiology Spectrum. Your manuscript has been reviewed by two experts in the field and both were enthusiastic about your work. Both reviewers had minor comments for you to address. When submitting the revised version of your paper, please provide (1) point-by-point responses to the issues raised by the reviewers as file type "Response to Reviewers," not in your cover letter, and (2) a PDF file that indicates the changes from the original submission (by highlighting or underlining the changes) as file type "Marked Up Manuscript - For Review Only". Please use this link to submit your revised manuscript - we strongly recommend that you submit your paper within the next 60 days or reach out to me. Detailed instructions on submitting your revised paper are below.

Link Not Available

Sincerely,

Philip Rather

Journals Department
Reviewer comments:

Reviewer #1 (Comments for the Author):

This interesting study by Maher et al., describes a TraDIS approach to investigating the effect of two structurally very similar antibiotics ampicillin and benzylpenicillin. The use of an unbiased screen to address how these compounds accumulate in E. coli is a strength of this study and has produced interesting results which will be of interest to the field. That said it is a description of a single experiment (with validation) so I think fits well into the observation format.

I appreciate that due to the article format there is no actual methods section but I think more methodological detail is required. For example, what growth phase were the cells grown to as this may well affect the genes that were detected.

In the introduction the authors state that this study investigated the importance of the primary amine group in promoting uptake. I think the final conclusions could be strengthened by returning to this posed question and commenting on what the data tells us about the impact of the amine.

Reviewer #2 (Comments for the Author):

In this study the authors used TraDIS to identify genes that affect E. coli fitness in the presence of sub-inhibitory concentrations of ampicillin or benzylpenicillin. It is important to understand the mechanisms that cause resistance to antibiotics in order to design methods to overcome these resistance mechanisms. This study is therefore timely and important.

In general, mutations in the outer membrane, particularly the lipopolysaccharide layer decreased the fitness under benzylpenicillin exposure. This result is in line with the notion of benzylpenicillin not being an effective antibiotic against Gram-negative organisms due to its inability to pass through outer membrane porins such as OmpF.

Some of the other findings could have been explored a bit more for example the effect of mutations in dacA.

Line 176 and onwards: The argument about the PEN being transported at a higher rate compared to AMP and the relationship between the inner membrane permeability and mutations in the regulatory machinery of AcrAB-TolC, needs to be better explained.

There is a few minor corrections needed for example:

Line 19 - Should this sentence read "...components of the envelope that prevent...."

Line 81 - E. coli not E.coli

Staff Comments:

Preparing Revision Guidelines

Please return the manuscript within 60 days; if you cannot complete the modification within this time period, please contact me. If you do not wish to modify the manuscript and prefer to submit it to another journal, please notify me of your decision immediately so that the manuscript may be formally withdrawn from consideration by Microbiology Spectrum.

Reviewer comments:

Reviewer #1 (Comments for the Author):

This interesting study by Maher et al., describes a TraDIS approach to investigating the effect of two structurally very similar antibiotics ampicillin and benzylpenicillin. The use of an unbiased screen to address how these compounds accumulate in *E. coli* is a strength of this study and has produced interesting results which will be of interest to the field. That said it is a description of a single experiment (with validation) so I think fits well into the observation format.

I appreciate that due to the article format there is no actual methods section but I think more methodological detail is required. For example, what growth phase were the cells grown to as this may well affect the genes that were detected.

Authors' response: Additional detail about library challenge methodology was added to the revised manuscript.

Lines 87–89 *"Cells were sampled at stationary phase to ensure the effects of compound accumulation throughout the cell cycle, including during active growth phases, were reflected in mutant abundance (10)".*

In the introduction the authors state that this study investigated the importance of the primary amine group in promoting uptake. I think the final conclusions could be strengthened by returning to this posed question and commenting on what the data tells us about the impact of the amine.

Authors' response: The potential role of the primary amine in ampicillin is addressed more explicitly in the conclusions.

Lines 213–216 *"Previous investigations into the role of the primary amine in ampicillin accumulation in Gram-negatives have primarily focussed on how this chemical feature affects compound uptake. The significance of this altered uptake efficiency in improving accumulation is supported by this study".*

Lines 218–220 *"However, the potential effects of the primary amine on compound efflux and the contribution of this to changes in overall levels of accumulation have been understudied".*

Reviewer #2 (Comments for the Author):

In this study the authors used TraDIS to identify genes that affect *E. coli* fitness in the presence of sub-inhibitory concentrations of ampicillin or benzylpenicillin. It is important to understand the mechanisms that cause resistance to antibiotics in order to design methods to overcome these resistance mechanisms. This study is therefore timely and important.

In general, mutations in the outer membrane, particularly the lipopolysaccharide layer decreased the fitness under benzylpenicillin exposure. This result is in line with the notion of benzylpenicillin not being an effective antibiotic against Gram-negative organisms due to its inability to pass through outer membrane porins such as OmpF.

Some of the other findings could have been explored a bit more for example the effect of mutations in *dacA*.

Authors' response: Further commentary on the effect of mutations in *dacA* has been added to the revised manuscript.

Lines 119–122 *"It has been suggested that PBP5, a redundant PBP, may act to sequester β -lactams away from essential PBPs, providing a shielding effect (12). This may explain why insertions in this PBP in particular were so detrimental to fitness under both treatments"*.

Line 176 and onwards: The argument about the PEN being transported at a higher rate compared to AMP and the relationship between the inner membrane permeability and mutations in the regulatory machinery of AcrAB-TolC, needs to be better explained.

Authors' response: This paragraph has been reworded to improve clarity.

Lines 176–189 *"Previous work has suggested that PEN may be a better substrate of AcrAB-TolC in *E. coli* than AMP (20). This may explain why AMP would impose higher selective pressure for the inactivation of the *acrR* repressor gene and maintenance of the *marA* activator gene than PEN, as a higher level of AcrAB expression may compensate for lower levels of AMP transport by this pump. Alternatively, PEN may be better able to promote *acrAB* expression than AMP and thus reduce selective pressure for mutations that increase its expression. Indeed, PEN has a higher membrane permeability than AMP and thus greater potential to diffuse into the cytoplasm where it could interact with cytoplasmic regulators like AcrR to relieve translational repression of *acrAB* (20, 21). To investigate this possibility, qRT-PCR was performed to examine *acrAB* expression in wild-type cells treated with AMP and PEN at concentrations equivalent to those used in this assay (22). These experiments did not show a clear change in expression for these genes for either antibiotic treatment (data not shown). Another possible reason for selective pressure on *acrR* and *marA* under AMP selection could be the roles of these regulators in controlling other genes, such as *ompF* (see below)"*.

There is a few minor corrections needed for example:

Line 19 - Should this sentence read "...components of the envelope that prevent....."

Authors' response: Correction made (line 19).

Line 81 - E. coli not E.coli

Authors' response: Correction made (line 80).

November 8, 2022

Dr. Karl A Hassan
University of Newcastle Australia
Newcastle, NSW 2308
Australia

Re: Spectrum03593-22R1 (Breaching the barrier: genome-wide investigation into the role of a primary amine in promoting E. coli outer-membrane passage and growth inhibition by ampicillin)

Dear Karl,

Your manuscript has been accepted, and I am forwarding it to the ASM Journals Department for publication. You will be notified when your proofs are ready to be viewed.

Sincerely,

Philip Rather
Editor, Microbiology Spectrum

Journals Department
Supplemental Dataset: Accept
Supplemental Material: Accept